# The Longitudinal Effect of Meditation on Resting-State Functional Connectivity Using Dynamic Arterial Spin Labeling: A Feasibility Study

**DOI:** 10.3390/brainsci11101263

**Published:** 2021-09-24

**Authors:** Zongpai Zhang, Wen-Ming Luh, Wenna Duan, Tony D. Zhou, Li Zhao, George Weinschenk, Adam K. Anderson, Weiying Dai

**Affiliations:** 1Department of Computer Science, State University of New York at Binghamton, Binghamton, NY 13902, USA; zzhan145@binghamton.edu (Z.Z.); wduan1@binghamton.edu (W.D.); tonyzhou.mp@gmail.com (T.D.Z.); weinscgg@binghamton.edu (G.W.); 2National Institute on Aging, National Institutes of Health, Baltimore, MD 21225, USA; wenming.luh@nih.gov; 3College of Biomedical Engineering & Instrument Science, Zhejiang University, Hangzhou 310027, China; lz5bf@virginia.edu; 4Department of Human Development, Cornell University, Ithaca, NY 14853, USA; aka47@cornell.edu

**Keywords:** meditation, functional magnetic resonance imaging, functional connectivity, arterial spin labeling, default mode network, dorsal attention network

## Abstract

We aimed to assess whether dynamic arterial spin labeling (dASL), a novel quantitative MRI technique with minimal contamination of subject motion and physiological noises, could detect the longitudinal effect of focused attention meditation (FAM) on resting-state functional connectivity (rsFC). A total of 10 novice meditators who recorded their FAM practice time were scanned at baseline and at the 2-month follow-up. Two-month meditation practice caused significantly increased rsFC between the left medial temporal (LMT) seed and precuneus area and between the right frontal eye (RFE) seed and medial prefrontal cortex. Meditation practice time was found to be positively associated with longitudinal changes of rsFC between the default mode network (DMN) and dorsal attention network (DAN), between DMN and insula, and between DAN and the frontoparietal control network (FPN) but negatively associated with changes of rsFC between DMN and FPN, and between DAN and visual regions. These findings demonstrate the capability of dASL in identifying the FAM-induced rsFC changes and suggest that the practice of FAM can strengthen the efficient control of FPN on fast switching between DMN and DAN and enhance the utilization of attentional resources with reduced focus on visual processing.

## 1. Introduction

Meditation provides evidence of a diverse set of benefits, including emotional regulation [1], awareness and self-regulation [2], memory and cognition [3], attention [4], and working memory [5]. Meditation has also been demonstrated to be beneficial to the treatment of psychological disorders [6,7,8,9]. Neuroimaging studies have examined the state effect and the trait effect of meditation. The state effect refers to the short-term consequences during an individual’s meditative practice. The trait effect characterizes the long-lasting changes that accrue via repetitive practice even when the meditation is not practiced. The therapeutic effects and daily life benefits (e.g., improved attention) emerge from the trait-related brain changes that are induced by meditation.

Meditation has been associated with trait changes of the brain structure using cross-sectional structural magnetic resonance imaging (sMRI) studies. Increased gray matter (GM) density [10,11,12] and cortical thickness [13,14,15,16] have been found in the default mode network (DMN) and dorsal attention network (DAN) regions in charge of self-referential and attention processing: such as superior parietal gyri (a posterior region in DAN), precuneus (a posterior region in DMN), anterior cingulate, superior and middle frontal gyri, and orbitofrontal (prefrontal regions in DMN) regions in meditators. Longitudinal sMRI studies further confirmed causal effects of meditation on increased GM density after an 8-week meditation training in the posterior cingulate cortex and temporoparietal junction (posterior regions in DMN) [15,17,18]. Therefore, we hypothesized that meditation would result in altered brain resting-state functional connectivity (rsFC) with key regions of DMN (posterior cingulate cortex (PCC) and ventromedial prefrontal cortex (vmPFC)) and DAN (left middle temporal (LMT), right middle temporal (RMT), left frontal eye (LFE), right frontal eye (RFE), left superior parietal lobule (LSPL), and right superior parietal lobule (RSPL)) [19] and that the changes are correlated with meditation practice time. In a recent study on the same subjects used in the present work, we have found that meditation can significantly increase rsFC between DMN and DAN [20].

The trait-related brain changes have been studied for the neural basis underlying the experience and training predominantly using blood-oxygen-level-dependent (BOLD) resting-state functional MRI (rsfMRI) [21]. Recent studies focused on understanding the effects of meditation on patterns of rsFC across distributed brain regions or networks (for a review, see [22]). The studies have provided ample evidence that meditation training alters brain rsFC and the changes of rsFC are associated with the duration of meditation experience. However, the recent literature review indicated few converging findings on the meditation effects on functional connectivity (FC) [22].

Pseudo-continuous arterial spin labeling (PCASL) [23] provides a noninvasive way of measuring three-dimensional (3D) cerebral blood flow (CBF). Recently, dynamic arterial spin labeling (dASL) [24], which measures rsFC by collecting a time series of CBF images, can detect brain resting-state networks without artifactual “networks” that are typically detected in BOLD studies prior to subjective filtering. BOLD rsfMRI signals consist of motion artifacts and physiological noises, including head motion, cardiac pulsation, and respiratory motion [25,26]. For example, increased head motion can increase the long-distance rsFC but decrease short-distance rsFC. It is possible that experienced meditators may exhibit less head motion within the scanner and slower cardiac and respiratory rates, especially during some meditation tasks, and the acquisition method should therefore account for possible differences from the factors. By contrast, dASL has the advantage of minimal influence from small motion artifacts and physiological noises because it can be combined with the other background suppression techniques to suppress the background tissue signals. dASL is sensitive to subject motion due to the subtraction of label and control images. Motion-induced static tissue signal changes cause imperfect subtraction and hence motion artifacts in the ASL difference images. Strong background suppression markedly reduces static tissue signals and therefore motion artifacts (i.e., motion-induced subtraction errors). This advantage is particularly relevant to meditation researches as frequent changes in the peripheral physiology of respiration and heart rate can markedly affect the detection of fMRI signals and estimates of functional connectivity. In addition, we have demonstrated that dASL with 3D fast spin-echo (FSE) image acquisition has minimal air-tissue susceptibility artifacts in areas such as the orbitofrontal cortex (OFC) and medial/inferior temporal cortex [24]. Hence, the background suppression of PCASL time series with 3D FSE readout can provide better sensitivity for probing rsFC with the susceptibility regions. Therefore, dASL is expected to be more sensitive in detecting the rsFC changes caused by meditation practice. The dASL time-series acquisition can quantify time-averaged CBF in an absolute physiological unit (mL/100 g·min) besides the measurement of rsFC. Additionally, it is unclear whether the trait-related changes are actually caused by the duration of meditation practice despite the reported association of meditation practice time with rsFC between brain regions or networks [27,28,29,30]. These findings may be caused by not accounting for baseline rsFC before performing the meditation. Herein, we chose a longitudinal study (to remove the effect of baseline rsFC) to investigate the feasibility and sensitivity of dASL in detecting the effects of meditation on rsFC and CBF. Considering the regulatory role of frontoparietal control network (FPN) in the attention control in response to salient stimuli [31,32] and consistent finding for inter-network connections among DMN, FPN, and salience network (SN) during active meditation state [33,34,35,36], we anticipated that dASL is capable of detecting rsFC with FPN and SN from DMN and DAN seeds. Given that there are many different forms of meditation, such as focused attention or open monitoring or non-dual awareness, we studied one specific type of meditation—focused attention meditation (FAM)—in the study to avoid potentially canceling effects from different meditation categories.

## 2. Methods

### 2.1. Sample Size Justification

Based on the increased PCC rsFC [37] after weeks of meditation training, we have previously derived an effect size of 0.81 with conventional BOLD [20]. Our dASL method increases the effect size by 49% for the ACC rsFC in bipolar disorder patients, compared to conventional BOLD fMRI [38]. However, the effect size of rsFC is dependent on the seed locations. Assuming that the maximum increase in the effect size for ACC rsFC is reached for the dASL method, we can consider the effect size of dASL increases by about half of 49% (i.e., 24%) on average from conventional BOLD. To detect a meditation effect size of 1.00 using dASL (24% of the expected increase in the effect size with the dASL method from the effect size of 0.81 with conventional BOLD) between the baseline and follow-up, a sample size of 9 is required to achieve a power of 80% and a level of significance of 5%. This study had an actual sample size of 10, which provides sufficient power to detect the meditation effect using the dASL method.

### 2.2. Study Population

A total of 11 healthy undergraduates at the Binghamton University (5 females, mean age: 19.09 ± 0.54 years, age range: 19 to 20 years) participated in the study. All participants were recruited from a university-offered meditation course. Nine participants had no prior meditation experience before, while two participants had a brief meditation period without any specific guidance in their prior yoga classes.

The meditation course introduced several different types of meditation techniques and reviewed the benefit of meditation on behaviors from literature. The instructor guided students for 15 min meditation once or twice per week during class in order to have students familiar with different meditation styles. Participants were instructed during class to sit comfortably, relax shoulders, close/open eyes, concentrate on a selected point focus (e.g., breath), and repeatedly bring the attention back to the selected focus if they noticed their attention had drifted. Participants could choose their own point of focus during the meditation practice, such as their breath, a spot on the wall, a phrase, or anything else they felt appropriate. Homework required practice FAM for at least 10 min per session and no less than 5 times per week and writing a weekly journal by describing their practice experiences.

All 11 participants underwent the baseline MRI scans before any homework assignments. One subject was not able to attend the follow-up scan after 2 months. Participants reported their total practice time according to their daily logs between the baseline and follow-up scans. The total practice time was calculated as the sum of all meditation practice time, including those from class and homework practice.

### 2.3. MRI Acquisition

We scanned meditation participants, including 11 at baseline and 10 at the 2-month follow-up), on a 3 Tesla GE MR750 scanner (General Electric, Milwaukee, WI, USA) using a 32-channel receive-only phased-array head coil at Cornell University MRI Facility. The Institutional Review Board of Cornell University approved the study, and all participants fully understood and signed written informed consent. All methods described in this manuscript were performed in strict accordance with the approved guidelines. The scans included 3D whole-brain sagittal T1-weighted magnetization prepared rapid gradient echo (MPRAGE) images for image registration, and the resting-state 3D dynamic arterial spin labeling (dASL), specifically, MPRAGE images were acquired in 5 min 30 s with the following parameters: field of view (FOV): 25 cm, 176 slices with matrix size: 256 × 256; slice thickness: 1.0 mm, repetition time (TR): 7 ms, echo time (TE): 3.42 ms, inversion time (TI): 425 ms, flip angle: 70, receiver bandwidth (rBW): 25 kHz. The pseudo-continuous arterial spin labeling (PCASL) sequence [24] was used to label the blood. The labeling duration was 2 s, with a delay of 1.8 s after labeling. Optimized background suppression pulses achieved less than 0.3% of background signals [24,39]. We acquired 50 3D dASL whole-brain volumes and a reference volume in 17 min. The reference volume was used for perfusion quantification. Each ASL volume, requiring a control-label pair and two spiral interleaves, was obtained with a 3D stack of spirals RARE sequence in a duration of 20 s.

### 2.4. Image Processing

#### 2.4.1. Quantification of Absolute CBF Maps

Reconstruction of the ASL label-control difference image time series was obtained by our custom reconstruction algorithm [24,40]. The first 3D ASL difference image was removed to increase stability for further processing. The head motion was corrected by realigning the remaining ASL image time series after which the average of the head-motion corrected images were generated. The average absolute CBF map for each subject was quantified using the standard kinetic model [41,42,43] with the mean of the head-motion-corrected ASL images and the reference image.

#### 2.4.2. Normalization of dASL Image Time Series and CBF Maps

For each subject, the motion-corrected ASL image time series were transformed to the standard MNI brain space. Specifically, GM, white matter, and cerebrospinal fluid probability maps were generated by segmenting high-resolution MPRAGE images. The GM probability maps were co-registered to the average of motion-corrected ASL images. The motion-corrected ASL time series and the average CBF map for each subject were transformed into the standard space by applying the same transformation that was obtained by the normalization from the co-registered GM probability map to the prior GM template in standard brain space. To correct for global CBF differences in the longitudinal setting, the relative CBF map for each subject was calculated by dividing the global mean CBF signal. The relative CBF was used because it was reported to be more sensitive in statistical analyses for group comparisons [44].

#### 2.4.3. Quantification of CBF Functional Connectivity Maps

Due to the minimal contamination from motion artifacts and physiological noises, the dASL image time series were not processed for removing the noises. Global CBF values, averaged over the whole brain, were regressed out from the dASL image time series. Eight seeds were chosen from the default mode network (DMN) and dorsal attention network (DAN). Two seeds were from DMN: the posterior cingulate cortex (PCC) and the ventromedial prefrontal cortex (vmPFC). Six seeds were from DAN: the left middle temporal area (LMT), the right middle temporal area (RMT), the left frontal eye field (LFE), the right frontal eye field (RFE), the left superior parietal lobule (LSPL), and the right superior parietal lobule (RSPL). The seeds were chosen to cover different anatomical extents: frontal cortex, parietal cortex, and temporal cortex. All seed regions of interest (ROIs) were defined as a sphere with a volume of ~2 cm^3^. The centers of the seed ROIs were selected from a previous study [19] (see a summary in Figure 1) that examined the spatial distribution of DMN and DAN. For each subject, a CBF rsFC map with a seed ROI was calculated as voxel-wise Pearson correlation coefficients between the dASL time series and the time series of each seed ROI. All Individual CBF rsFC maps with all seed ROIs were transformed into z score maps by using a Fisher z transformation to improve normality for group-level comparisons.

### 2.5. Statistical Analysis

The relative CBF maps and z-score maps from all seed ROIs in the standard space were smoothed with a 6 × 6 × 6 mm^3^ Gaussian kernel. In order to evaluate the changes of CBF measures from the baseline to 2-month follow-up, the z score maps, and relative CBF maps were modeled, respectively, using SPM12 via general linear model (GLM). In the GLM, the z score maps from each seed (and the relative CBF maps) were used as the dependent variables. The status index (baseline or follow-up), subject index (whether two scans are from the same subject or not), and gender were used as the independent variables. Age was not used as a covariate because the participants had a maximum of a 1-year age difference. The voxel-level significance threshold was set to *p* < 0.005. A cluster-level *p*-value of 0.05 was used to correct for multiple comparisons that were controlled by family-wise error (FWE). A more liberal voxel-level *p*-value threshold of 0.01 and FWE-corrected cluster-level *p*-value threshold of 0.05 were also used to view the trend if the above stringent *p* values failed to detect the longitudinal effects of meditation. The automated anatomical labeling (AAL) atlas binary mask was used to analyze only the GM regions.

To investigate the association of the longitudinal changes in CBF measures with meditation practice time, the difference of the z-score maps and relative CBF maps between the baseline and follow-up was modeled using SPM12 via GLM. In GLM, the difference maps from each contrast were used as the dependent variables. Gender and practice time were used as independent variables. The voxel-level *p*-value and FWE-corrected cluster-level *p*-value thresholds were set as 0.005 (more liberal value of 0.01 later to view the trend) and 0.05.

Post hoc regional analyses were performed to visualize the longitudinal changes and the relationship between the longitudinal changes and meditation practice time in those CBF measures. The significant clusters obtained from either voxel-based analysis were separated into different ROIs. The regional value for each ROI was calculated as the mean signal value over all the voxels within the region. Post hoc regional analyses were performed using paired *t*-test with gender as a covariate and multiple linear regression analysis with gender and meditation practice time as covariates, respectively.

## 3. Results

### 3.1. Basic Characteristics of the Participants

Table 1 summarizes the participants’ information and meditation practice time. One participant was not scanned at follow-up. The meditation practice duration and practice time for the remaining ten participants is 66.50 ± 4.14 days and 574.00 ± 465.55 min. Only 10 participants were included in the longitudinal analysis. Among them, two had practice time of more than 1000 min.

### 3.2. Quality of dASL Time Series

dASL image time series was of good quality. Representative dASL images from four random time points and their temporal signal-to-noise ratio (SNR) map across all time points are shown in Appendix A. The temporal SNR map for each subject was calculated as the mean dASL image divided by the standard deviation map across all time points. The GM temporal SNR value for each dASL scan was calculated by averaging the temporal SNR map over the GM ROI. GM ROI was defined as the voxels that have over 60% of probability in the SPM12 tissue probability map. The average temporal SNR value across 10 subjects was 7.83 ± 1.19 at the baseline and 7.65 ± 0.97 at follow-up.

### 3.3. Differences in Global CBF and Relative CBF between the Baseline and Follow-Up

No significant global CBF changes (*p* = 0.52) were found after correcting for gender effects from the baseline to follow-up. No significant relative CBF changes were observed from the baseline to follow-up using a voxel-level *p* < 0.005. With a liberal voxel-level *p* < 0.01, we observed a relative CBF increase in the junction of occipital, temporal and parietal regions (corrected cluster-level *p* = 0.027) (Appendix A) and a relative CBF decrease primarily in the thalamus region (corrected cluster-level *p* = 0.002) (Appendix A) at follow-up, compared to the baseline. A summary of the two clusters’ statistics is listed in Appendix A.

### 3.4. Association of Relative CBF with Meditation Practice Time

No significant relative CBF changes were observed to associate with meditation practice time from the baseline to follow-up using a voxel-level *p* < 0.005. With a liberal voxel-level *p* < 0.01, we observed that more practice time was associated with less longitudinal CBF changes in the junction of occipital, temporal, and cerebellum regions (Appendix A). A summary of the two clusters’ statistics is listed in Appendix A. Regional analysis showed that the initially increased CBF gradually reduced with practice time (Appendix A). The two subjects with practice time longer than 1000 min exhibited reduced CBF at follow-up, compared to their baseline CBF values. It is worth noting that the region with mixed CBF increase and decrease from different subjects, is very close to the region with longitudinal CBF increases.

### 3.5. Differences in CBF Functional Connectivity between Baseline and Follow-Up

Significant longitudinal changes in CBF functional connectivity were observed from the baseline to follow-up using a voxel-level *p* < 0.005. Compared to the baseline, we observed significantly increased CBF rsFC between the LMT seed and precuneus area (corrected cluster-level *p* = 0.023) (Figure 2a) and between the RFE seed and the superior frontal area (corrected cluster-level *p* < 0.001) (Figure 2b) and decreased CBF rsFC between the LMT and the temporal area (corrected cluster-level *p* = 0.012) (Figure 2c) at follow-up. A summary of the clusters’ statistics is reported in Table 2. Regional analysis showed the rsFC change from the baseline to follow-up with different seeds (Figure 3).

### 3.6. Association of CBF Functional Connectivity with Meditation Practice Time

Longitudinal changes in CBF functional connectivity with the DMN seeds were found to correlate with meditation practice time using a voxel-level *p* < 0.005. After adjusting for the gender effect, the practice time was positively associated with the longitudinal changes in rsFC between the PCC seed and the insular and prefrontal areas (Figure 4a), between the vmPFC seed and the frontal and temporal region (Figure 4c), but negatively associated with the longitudinal changes in rsFC between the PCC seed and the precuneus area (Figure 4b) and between the vmPFC seed and the frontal and parietal area (Figure 4d). A summary of the clusters’ statistics is reported in Table 3.

Longitudinal changes in CBF functional connectivity with the DAN seeds were found to correlate with meditation practice time using a voxel-level *p* < 0.005. After adjusting for the gender effect, the practice time was found to be positively associated with the longitudinal changes in rsFC between the RMT seed and the middle/inferior frontal area (Figure 5a), between the LSPL seed and the middle/inferior frontal area (Figure 5c), between the RFE seed and the superior/middle frontal area (Figure 5e), and between the LFE seed and the frontal and angular/parietal area (Figure 5g). The meditation practice time was found to be negatively associated with the longitudinal changes in rsFC between the RMT seed and the occipitoparietal area (Figure 5b), between the LSPL seed and the occipitoparietal area (Figure 5d), and between the RFE seed and the anterior/middle cingulate area (Figure 5f). A summary of the clusters’ statistics is also reported in Table 3. Post hoc regional analysis showed that the longitudinal changes of rsFC may change from a positive to a negative direction (increased rsFC to decreased rsFC at follow-up compared to the baseline) (Appendix A) or from negative to positive (decreased rsFC to increased rsFC at follow-up, compared to the baseline) (Appendix A) as more meditation practice time was involved. We also noticed that the switching time between different polarities is about 500 to 700 min of meditation practice time. A complete summary of post hoc regional correlation with total medication practice time is listed in Appendix A.

## 4. Discussion

In this longitudinal study, altered rsFC were found following 2-month meditation practice using the dASL technique. The LMT rsFC significantly increased in the precuneus region but decreased in the superior temporal region; the RFE rsFC significantly increased in the superior frontal region. Total meditation practice time was positively associated with rsFC between PCC and insular/temporal/mPFC, vmPFC and frontal/temporal/precuneus, RMT and middle/inferior frontal, LSPL/RSPL and left middle/inferior frontal, RFE and superior/middle frontal, LFE and frontal and angular/parietal regions. Total meditation practice time was negatively associated with rsFC between PCC and precuneus/occipital, vmPFC and frontal and parietal, RMT and occipitoparietal, LSPL and left occipitoparietal, and RFE and anterior/middle cingulate regions.

The longitudinal increase in rsFC between the LMT and precuneus and between RFE and superior frontal regions expands the current literature that reports the stronger rsFC between certain regions of the DMN and certain regions of the DAN [33,45]. These results are also consistent with the enhanced DMN-DAN rsFC using the BOLD fMRI technique and advanced denoising technique. The strengthened DMN-DAN rsFC suggests greater synergy coherence between self-referential and attention and may facilitate switching between the networks [46]. The enhanced coupling of DMN and DAN may reflect the benefits of meditation processes, such as improved attention regulation and self-monitoring.

Longer meditation practice time was associated with (1) increased PCC-rsFC and vmPFC-rsFC with the mPFC/ACC area and (2) increased PCC-rsFC with the insula area. These results extend the prior cross-sectional study that showed greater DMN-rsFC with the mPFC region [34] and PCC-rsFC with the ACC region [33] in experienced meditators than healthy controls. Cross-sectional studies also reported the increased mPFC-rsFC with the insula region in experienced meditators compared to controls [33,45] and increased salience network rsFC with the PCC region [47] in the high practice group, compared to the low practice group. Insula, which is the core region of the brain salience network, is thought to detect salient features for additional processing and act as a switchboard to direct other brain networks [48,49,50]. The direct causal influence of the insular cortex on PCC activity has been established [50], which provides further mechanistic evidence of the salience network exerting influence on DMN through meditation.

Meditation practice time was associated with increased dorsal lateral prefrontal cortex (DLPFC) rsFC with almost all the DAN seeds, including RMT, LSPL/RSPL, LFE/RFE. The DLPFC region, a key region in the frontoparietal control network (FPN), serves to initiate and regulate attention and cognitive control in response to salient stimuli [31,32]. Our results are in good accordance with the work by Taren et al. that showed mindfulness training increases rsFC between DLPFC and DAN (although they used the DLPFC as a seed for rsFC investigation) [51] and by Froeliger et al. that showed increased rsFC between FPN and DAN (specifically between DLPFC and inferior parietal regions) [30]. Consistent with its anatomical connections [32,52], the DLPFC region extends its rsFC to all DAN regions, instead of prior reported specific regions, indicating the higher sensitivity of our technique on rsFC. The increased rsFC between DLPFC and all DAN seeds suggests that FAM can enhance the top-down control ability for attention selection in the functionally connected DAN system. In addition, we have extended the correlation of practice time with DAN-FPN rsFC at a single time point [30] to the correlation with the changes of DAN-FPN rsFC. These longitudinal results demonstrated that the changes of DAN-FPN rsFC are caused by meditation practice.

Meditation practice time was associated with decreased vmPFC-rsFC with DLPFC. These findings are consistent with reduced rsFC between DMN and FPN/CEN for meditators [53,54]. Conversely, a recent study reported a positive correlation between meditation experience and mPFC-CEN rsFC [55], which seems to contradict our increased anticorrelation with practice time (Appendix A). However, their further analysis resolved the conflicts by noticing that the intermediate meditators (<1130 h of practice, approximately three years of 1 h daily practice) exhibited significant increases in anticorrelations between CEN and MPFC, whereas more experienced meditators (>1130 h of practice) have the anticorrelations returned to a premeditation state. Our longitudinal results, together with the cross-sectional results, further underscore that meditation changes brain connectivity in a nonlinear way. The increased anticorrelations between CEN and DMN suggest that CEN negatively regulates DMN (i.e., suppresses the brain activity of DMN), and thereby meditators probably spend less time in a mind-wandering state during daily life [56].

Meditation practice time was associated with decreased RMT-rsFC or LSPL/RSPL-rsFC with visual/occipital regions. These results are in line with lower meditative-state connectivity between attention and visual networks in more experienced meditators [47]. These results suggest that meditators may have a more efficient attentional allocation, with decreased attentional resources being devoted to the visual processing domain. In addition, we also observed a negative association between meditation practice time and RFE-ACC rsFC. ACC is anatomically connected with the frontal eye fields and assigns appropriate control to visual attention [57]. The reduced rsFC between ACC and RFE also indicates meditation exerting more efficient attentional control, with a decreased assignment to visual attention.

We found a significant negative correlation between PCC-precuneus/inferior parietal connectivity and meditation practice time. These results were a little surprising to us because we observed a positive correlation between PCC-mPFC and mPFC-mPFC connectivity and meditation practice time. However, these results are consistent with the anticorrelation in DMN (between nearby DMN regions: precuneus and inferior parietal regions in their findings) [29]. These findings are also in line with several studies that have found reduced connectivity within certain regions of the DMN in experienced meditators, compared to their novice counterparts [28,54,58].

We observed that the individual differences in meditation practice time were associated with rsFC in markedly more pairs of brain regions using the dASL technique compared with the multi-echo BOLD fMRI technique (results shown as a separate publication [20]) in the same participant sample. Meditation practice time was positively correlated with DMN-DAN rsFC changes using the multi-echo BOLD fMRI technique. By contrast, meditation practice time was positively correlated with DMN-DAN rsFC changes, and also positively correlated DMN-SN and DAN-FPN rsFC changes, and negatively correlated with DMN-FPN rsFC using the dASL technique. The extra rsFC findings using dASL reveal that FPN and SN are involved in brain attentional switching between DMN and DAN, and 2-month meditation practice can enhance the communication between those networks, reflecting the higher sensitivity of the dASL technique in characterizing the rsFC. We postulate that the higher sensitivity of dASL is because its signals are more robust to physiological noises and small motion artifacts via the incorporation of strong background suppression techniques. Strong background suppression has been proven useful in limiting physiological noises and motion-induced subtraction errors in renal ASL applications [59,60]. Meanwhile, we found fewer pairs of brain regions with longitudinal rsFC changes following 2-month meditation practice using the dASL technique. CBF rsFC changes from the baseline to follow-up experienced different polarities (some participants had increased rsFC but others had decreased rsFC from the baseline to follow-up) (see Appendix A). Different polarities in CBF rsFC across subjects in our study cancel each other out, and therefore, significant changes of CBF rsFC on group level were less prominent. We expect that the CBF rsFC using dASL will achieve superior sensitivity in detecting longitudinal changes when all participants have either increased rsFC or decreased rsFC. Increased rsFC or decreased rsFC across participants can be sensitively detected when all participants practice meditation longer than the zero-crossing time (e.g., ~750 min for PFC-parietal rsFC in Appendix A). The polarity differences in longitudinal rsFC changes between dASL and multi-echo BOLD presumably emerge from their different signal sources. dASL signals are from a single source CBF, while BOLD signals reflect a combination of effects from blood oxygenation, cerebral blood volume, CBF, and metabolic rate of oxygen changes [61,62,63,64]. BOLD signal fluctuations from sources other than CBF may have contributed to the polarity difference, i.e., different signs of rsFC changes, compared to dASL.

This study has some limitations. First, the study has a small sample size. However, the study was intended to serve as a preliminary study to test the sensitivity of dASL in meditation effects. The observed significant rsFC changes after a 2-month meditation practice with this small sample size support the large effect size from the dASL method. Second, a control group was not included in the study. However, the longitudinal changes of rsFC were related to meditation practice time, suggesting that the rsFC changes emerge from meditation practice itself, not from expectancy bias. Third, a short follow-up period and only one follow-up time were used in the study. Therefore, the results may not be generalized to the effect of long-term meditation. Further investigation with a larger sample size, matched control group and longer follow-up is warranted to validate the effect of meditation practice time on brain rsFC changes.

## 5. Conclusions

Our significant findings using dASL support further investigation into the power of the technique in meditation studies. Two-month meditation is associated with the changes of rsFC and the changes of rsFC are associated with practice time. These findings suggest that the practice of meditation can strengthen the efficient control of FPN on fast switching between DMN and DAN and improve the utilization of attentional resources with reduced focus on visual processing.

## Figures and Tables

**Figure 1 brainsci-11-01263-f001:**
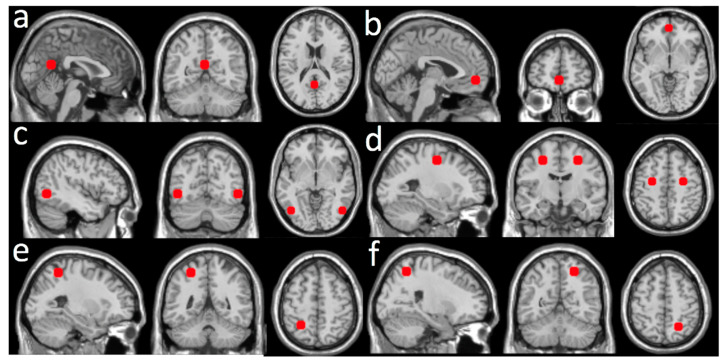
ROIs constructed from the seeds with the MNI coordinates listed: (**a**) posterior cingulate cortex (PCC, (1, −55, 17) mm); (**b**) ventromedial prefrontal cortex (PFC, (0, 51, −7) mm); (**c**) left and right middle temporal area (LMT, (−45, −69, −2) mm and RMT, (50, −69, −3) mm); (**d**) left and right frontal eye fields (LFE, (−25, −8, 50) mm and RFE, (27, −8, 50) mm); (**e**) left superior parietal lobule (LSPL, (−27, −52, 57) mm); (**f**) right superior parietal lobule (RSPL, (24, −56, 55) mm).

**Figure 2 brainsci-11-01263-f002:**
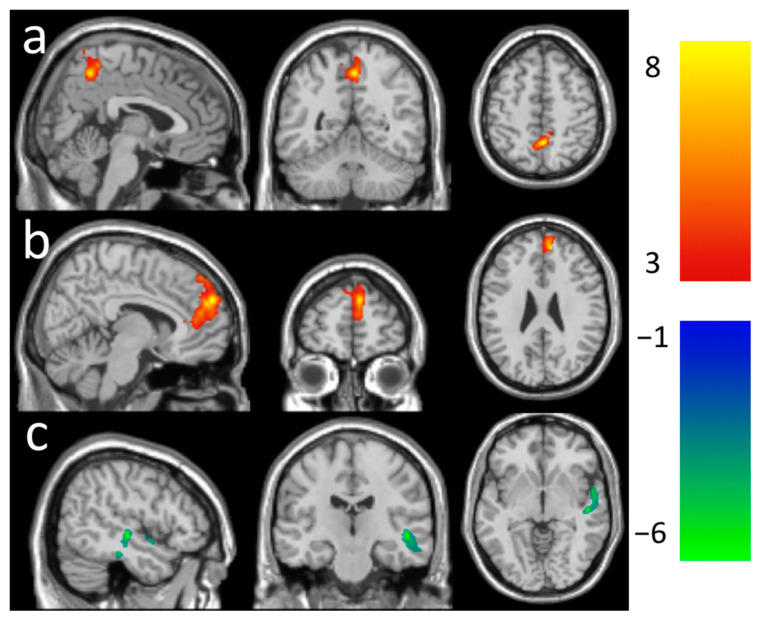
Increased functional connectivity (**a**) in the precuneus area (peak-T at 4, −52, 52) with the left visual region seed ROI (LMT) and (**b**) in the frontal region (peak-T at (10, 26, 46)) with the right frontal eye field seed ROI (RFE). (**c**) Decreased functional connectivity in the superior temporal area (peak-T at (50, −20, −6)) with the left visual region seed ROI (LMT) after 2-month meditation practice at the thresholds of a voxel-level *p*-value of 0.005 and corrected cluster-level *p*-value of 0.05. The color bar shows the range of *t*-values.

**Figure 3 brainsci-11-01263-f003:**
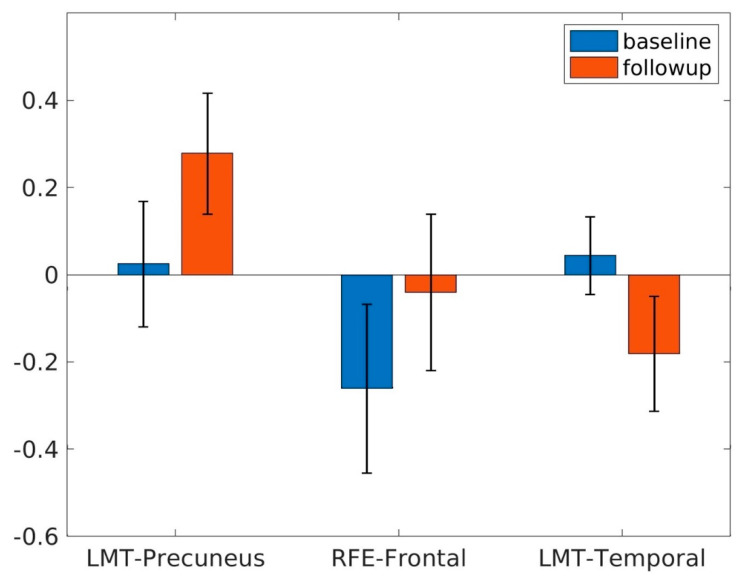
Functional connectivity between the LMT and precuneus, RFE and frontal, and LMT and temporal regions at baseline and follow-up.

**Figure 4 brainsci-11-01263-f004:**
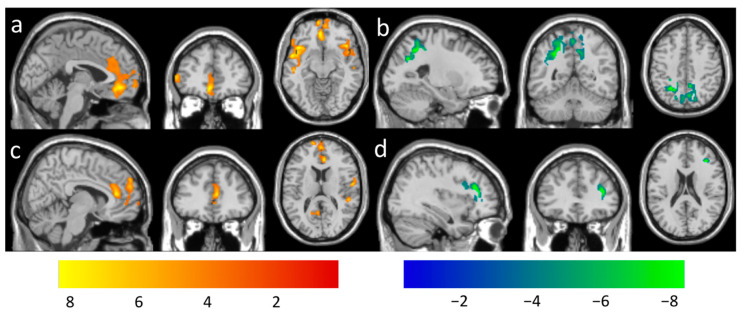
Regions overlaid on a standard brain template in which the meditation practice time was positively associated with functional connectivity with a seed ROI at the (**a**) PCC (peak-T −54, 4, −10), (**c**) vmPFC (peak-T 8, 36, 14) and negatively associated with functional connectivity with a seed ROI at the (**b**) PCC (peak-T −26, −52, 42), (**d**) vmPFC (peak-T 36, 34, 18) at the thresholds of a voxel-level *p*-value of 0.005 and corrected cluster-level *p*-value of 0.05. The color bar shows the range of *t*-values.

**Figure 5 brainsci-11-01263-f005:**
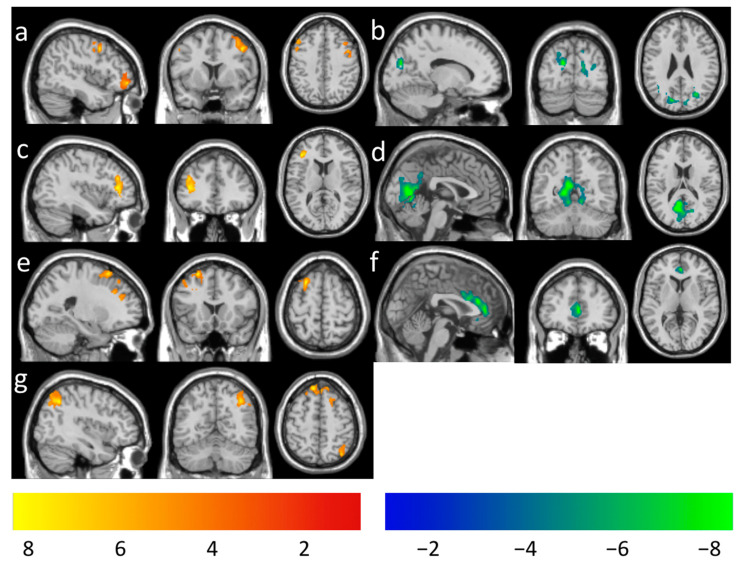
Regions overlaid on a standard brain template in which the meditation practice time was positively associated with functional connectivity with a seed ROI at the (**a**) RMT (peak-T 44, 8, 44), (**c**) LSPL (peak-T −38, 34, 12), and (**e**) RFE (peak-T 24, 18, 60), (**g**) LFE (peak-T 40, −56, 46) and negatively associated with functional connectivity with a seed ROI at the (**b**) RMT (peak-T 8, −68, 22), (**d**) LSPL (peak-T −2, −64, 4), and (**f**) RFE (peak-T 0, 32, 24) at the thresholds of a voxel-level *p*-value of 0.005 and corrected cluster-level *p*-value of 0.05. The color bar shows the range of *t*-values.

**Table 1 brainsci-11-01263-t001:** Summary of demographical characteristics and meditation practice time.

	At Follow-Up
Total number of included subjects *	10
Age range	19 years (8), 20 years (2)
Gender	Male (6), Female (4)
Handedness	Right (7), Left (3)
Total practice time (minutes)	Range: (135, 1620) 574 ± 465
Time duration (days)	Range: (45, 72) 66 ± 4

* One subject was excluded because of not being scanned at follow-up.

**Table 2 brainsci-11-01263-t002:** Summary of cluster-level statistics for clusters showing significant longitudinal rsFC changes between the baseline and the follow-up.

	N Voxels	Local Zscore	Peak-t MNI Coordinates	Mean Beta ± Std.	Anatomical Locations	% Cluster	% Region
Increased	440	4.14	4, −52, 52	0.25 ± 0.03	Parietal Lobe		
LMT rsFC (Figure 2a)					Precuneus_R	60.68	8.18
					Precuneus_L	34.77	4.34
					Paracentral_Lobule_R	2.73	1.44
					Limbic System		
					Cingulum_Mid_R	1.82	0.36
Increased	1038	4.36	10, 50, 26	0.22 ± 0.04	Frontal Lobe		
RFE rsFC (Figure 2b)					Frontal_Sup_Medial_R	49.52	24.09
					Frontal_Sup_Medial_L	19.46	6.75
					Frontal_Sup_L	5.01	1.44
					Frontal_Sup_R	4.72	1.21
					Limbic System		
					Cingulum_Ant_R	20.81	16.45
Decreased	499	3.82	50, −20, −6	−0.23 ± 0.04	Temporal Lobe		
LMT rsFC (Figure 2c)					Temporal_Sup_R	48.30	7.67
					Temporal_Mid_R	24.65	2.79
					Temporal_Inf_R	16.63	2.33
					Temporal_Pole_Sup_R	6.81	2.54
					Heschl_R	2.00	4.02
					Temporal_Pole_Mid_R	1.00	0.42

MNI = Montreal Neurological Institute.

**Table 3 brainsci-11-01263-t003:** Summary of cluster-level statistics for clusters showing significant longitudinal FC changes associated with meditation practice time.

Association with Practice Time	N Voxels	Local Zscore	Peak-t MNI Coordinates	Mean Beta ± Std. (×10^−4^) (/minute)	Anatomical Locations	% Cluster	% Region
Positive for	2162	4.45	−54, 4, −10	6.45 ± 1.45	Insula		
PCC rsFC					Insula_L	22.48	26.16
(Figure 4a)					Temporal Lobe		
					Temporal_Pole_Sup_L	17.44	29.34
					Temporal_Sup_L	15.17	14.29
					Temporal_Mid_L	5.18	2.17
					Heschl_L	2.78	26.67
					Frontal Lobe		
					Frontal_Inf_Tri_L	14.52	12.42
					Frontal_Inf_Orb_L	11.05	14.14
					Frontal_Inf_Oper_L	5.64	11.75
					Rolandic_Oper_L	3.42	7.47
					Frontal_Mid_L	1.16	0.51
	1720	4.10	−4, −42, −12	7.14 ± 1.28	Limbic System		
					Cingulum_Ant_L	23.95	29.43
					Cingulum_Mid_L	3.84	3.40
					Cingulum_Ant_R	2.33	3.05
					Frontal Lobe		
					Frontal_Sup_Medial_L	17.44	10.03
					Frontal_Med_Orb_R	15.93	32.01
					Frontal_Med_Orb_L	15.81	37.83
					Rectus_L	6.63	13.38
					Frontal_Sup_Medial_R	6.63	5.34
					Rectus_R	4.59	10.60
					Frontal_Sup_L	2.21	1.06
	897	4.12	44, 22, −10	5.86 ± 1.32	Insula		
					Insula_R	34.23	17.34
					Temporal Lobe		
					Temporal_Pole_Sup_R	24.64	16.52
					Temporal_Pole_Mid_R	4.91	3.71
					Temporal_Mid_R	1.67	0.34
					Frontal Lobe		
					Frontal_Inf_Orb_R	16.50	8.67
					Frontal_Inf_Oper_R	13.60	8.72
					Frontal_Inf_Tri_R	3.68	1.53
	395	3.70	44, −24, 18	5.11 ± 0.86	Temporal Lobe		
					Temporal_Sup_R	42.53	5.35
					Heschl_R	14.68	23.29
					Temporal_Mid_R	1.52	0.14
					Frontal Lobe		
					Rolandic_Oper_R	30.13	8.94
					Insula		
					Insula_R	11.14	2.49
Negative for	2892	5.38	−26, −52, 42	−6.22 ± 1.38	Parietal Lobe		
PCC rsFC (Figure 4b)					Precuneus_L	26.38	21.63
					Precuneus_R	26.00	23.03
					Parietal_Sup_L	8.78	12.30
					Parietal_Inf_L	6.40	7.56
					Angular_L	3.63	8.95
					Parietal_Sup_R	3.56	4.64
					Postcentral_L	1.28	0.95
					Occipital Lobe		
					Occipital_Mid_R	11.45	15.78
					Occipital_Sup_R	4.88	9.98
					Occipital_Mid_L	3.67	3.24
					Occipital_Sup_L	1.38	2.93
					Cuneus_R	1.35	2.74
Positive with	1108	4.14	8, 36, 14	5.66 ± 1.11	Frontal Lobe		
vmPFC rsFC (Figure 4c)					Frontal_Sup_Medial_R	33.39	17.34
					Frontal_Sup_Medial_L	27.71	10.26
					Frontal_Sup_L	3.79	1.17
					Limbic System		
					Cingulum_Ant_R	27.8	23.46
					Cingulum_Ant_L	5.42	4.29
					Cingulum_Mid_R	1.17	0.59
	916	3.94	44, −26, 2	5.76 ± 1.03	Temporal Lobe		
					Temporal_Sup_R	61.68	17.99
					Temporal_Mid_R	9.28	1.93
					Temporal_Inf_R	5.02	1.29
					Heschl_R	2.84	10.44
					Frontal Lobe		
					Rolandic_Oper_R	11.90	8.19
					Precentral_R	1.20	0.33
					Parietal Lobe		
					Postcentral_R	8.08	1.94
	381	3.18	−2, −58, 18	6.27 ± 0.96	Parietal Lobe		
					Precuneus_L	45.14	4.88
					Precuneus_R	8.14	0.95
					Occipital Lobe		
					Calcarine_L	20.73	3.50
					Cuneus_L	9.71	2.42
					Lingual_L	1.84	0.33
					Limbic System		
					Cingulum_Post_L	11.02	9.07
					Cingulum_Mid_R	1.84	0.32
					Cingulum_Post_R	1.57	1.79
Negative for	377	5.16	36, 34, 18	−6.13 ± 1.31	Frontal Lobe		
vmPFC rsFC (Figure 4d)					Frontal_Inf_Tri_R	42.18	7.39
					Frontal_Mid_R	27.06	2.00
					Frontal_Inf_Oper_R	23.08	6.22
					Insula		
					Insula_R	7.69	1.64
	361	3.96	22, −68, 52	4.62 ± 0.86	Parietal Lobe		
					Parietal_Sup_R	72.85	11.84
					Angular_R	4.99	1.03
					Postcentral_R	1.94	0.18
					Occipital Lobe		
					Occipital_Sup_R	17.73	4.53
					Occipital_Mid_R	2.22	0.38
Positive for	596	3.78	44, 8, 44	5.79 ± 1.13	Frontal Lobe		
RMT rsFC (Figure 5a)					Frontal_Mid_R	62.58	7.31
					Precentral_R	26.01	4.58
					Frontal_Sup_R	11.41	1.68
	484	3.76	−50, 28, 28	6.33 ± 0.98	Frontal Lobe		
					Frontal_Inf_Tri_L	47.73	9.13
					Frontal_Mid_L	40.91	4.06
					Frontal_Inf_Oper_L	7.23	3.37
					Precentral_L	4.13	0.57
	330	4.34	46, 50, −8	5.69 ± 1.25	Frontal Lobe		
					Frontal_Inf_Orb_R	36.36	7.03
					Frontal_Mid_R	28.28	1.82
					Frontal_Mid_Orb_R	20.61	6.70
					Frontal_Inf_Tri_R	8.48	1.30
					Frontal_Sup_R	6.36	0.52
Negative for	724	3.89	28, −68, 22	−6.75 ± 1.40	Parietal Lobe		
RMT rsFC (Figure 5b)					Precuneus_R	23.07	5.11
					Parietal_Sup_R	8.29	2.70
					Occipital Lobe		
					Occipital_Mid_R	21.41	7.39
					Calcarine_R	16.44	6.39
					Occipital_Sup_R	15.88	8.14
					Cuneus_R	14.92	7.58
	571	4.29	−14, −78, 24	−6.59 ± 1.10	Occipital Lobe		
					Occipital_Sup_L	40.98	17.13
					Occipital_Mid_L	22.94	4.01
					Cuneus_L	11.38	4.26
					Calarine_L	3.33	0.84
					Parietal Lobe		
					Angular_L	16.99	8.27
					Parietal_Sup_L	1.40	0.39
					Temporal Lobe		
					Temporal_Mid_L	2.80	0.32
Positive for	403	3.47	−38, 34, 12	6.10 ± 1.26	Frontal Lobe		
LSPL rsFC (Figure 5c)					Frontal_Mid_L	52.85	4.38
					Frontal_Inf_Tri_L	40.94	6.52
					Frontal_Sup_L	6.20	0.69
Negative for	3681	5.55	−2, −64, 4	−6.49 ± 1.35	Occipital Lobe		
LSPL rsFC (Figure 5d)					Calcarine_L	24.67	40.21
					Lingual_L	16.25	28.54
					Lingual_R	14.34	22.96
					Calcarine_R	12.85	25.42
					Cuneus_L	7.23	17.43
					Parietal Lobe		
					Precuneus_L	9.05	9.44
					Precuneus_R	1.52	1.72
					Cerebelum		
					Cerebelum_4_5_L	3.15	10.31
					Vermis_4_5	1.98	10.98
					Cerebelum_4_5_R	1.33	5.69
					Vermis_6	1.09	10.78
					Limbic System		
					Cingulum_Post_L	2.50	19.87
Positive for	727	4.15	−24, 18, 60	−5.95 ± 1.29	Frontal Lobe		
RFE rsFC (Figure 5e)					Frontal_mid_L	51.03	7.63
					Frontal_Sup_L	33.70	6.81
					Frontal_Inf_Oper_L	7.57	5.30
					Precentral_L	4.68	0.96
					Frontal_Inf_Tri_L	3.03	0.87
Negative for	1010	4.33	0, 32, 24	5.68 ± 1.20	Limbic System		
RFE rsFC (Figure 5f)					Cingulum_Ant_L	56.63	40.86
					Cingulum_Mid_L	20.69	10.77
					Cingulum_Ant_R	25.54	11.96
					Cingulum_Mid_R	2.28	1.04
					Frontal Lobe		
					Frontal_sup_Medial_L	4.16	1.40
Positive for	546	3.92	40, −56, 46	5.84 ± 1.26	Parietal Lobe		
LFE rsFC (Figure 5g)					Angular_R	70.51	21.97
					Parietal_Inf_R	17.40	7.06
					Parietal_Sup_R	10.99	2.70
	543	4.39	26, 20, 64	5.26 ± 0.88	Frontal Lobe		
					Frontal_Sup_R	46.04	6.16
					Frontal_Sup_Medial_L	23.39	4.24
					Frontal_Sup_Medial_R	25.65	3.98
					Frontal_Sup_L	11.42	1.72
					Supp_Motor_Area_R	1.66	0.38
					Frontal_Mid_L	1.66	0.19

MNI = Montreal Neurological Institute.

## Data Availability

Raw data were generated from the MRI scanner. Reconstruction software is the vendor’s proprietary product. Sharing of derived data will be supported by direct request. After publishing our main findings, requests for data will be evaluated on a case-by-case basis. Before sharing data, we will make sure that all data are free of identifiers that could directly or indirectly link information to an individual and that all sharing is compliant with institutional and IRB policies.

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
