# Peer review of "The Longitudinal Effect of Meditation on Resting-State Functional Connectivity Using Dynamic Arterial Spin Labeling: A Feasibility Study"

_brainsci, 2021, doi:10.3390/brainsci11101263_

Round 1

Reviewer 1 Report

The authors used a novel quantitative MRI technique to evaluate longitudinal changes in functional connectivity between the default mode (DMN) and dorsal attention network (DAN) regions with the rest of the brain after 2 months of regular meditation practice. They found significant changes in functional connectivity after meditation practice in a sample of 10 individuals, as well as associations between meditation practice time and further functional connectivity differences, focused on the DMN and DAN.

Introduction

  • Please also introduce abbreviations at the beginning of introduction, even if they had already been introduced in the abstract.
  • Make sure you correct occasional spelling mistakes (especially “mediation” instead of “meditation”).
  • The hypothesis (line 49) is not very clear, please be more specific about the hypothesized seeds and targets of the analysis.

Methods

  • The chosen seeds seem reasonable. Nevertheless, it would be beneficial to add a bit more information on the reasons behind the choice of the seeds. E.g., why these regions, why these coordinates?
  • Statistical analysis: please also specify what type of a multiple comparison correction was used (e.g. FWE).

Supplementary Materials

  • Supplementary materials would benefit from clearer organisation, e.g. additional subtitles, to make it easy for the reader to find the information they are interested in.

Reviewer 2 Report

The authors performed cerebral blood flow (CBF) measurements with pseudo-continuous arterial spin labeling in novice meditation practitioners before and after 2 months’ focused attention meditation training. The rsFC was measured by CBF changes and was compared in different brain regions. The authors detected rsFC changes in a few brain regions in these subjects and identified some regions where the rsFC is correlated with meditation practice time. The idea of examining longitudinal changes associated with meditation is interesting. Overall, the experimental designs and general analysis were fine; however, I have serious concerns regarding “recycling” published texts and how the manuscript was constructed in general.

The authors have recently published a paper on the very same topic, using multi-echo BOLD fMRI instead of PCASL as the imaging tool: https://www.nature.com/articles/s41598-021-90729-y  (Received: 17 September 2020; Accepted: 31 March 2021).

It is understandable that the authors acquired ME BOLD and ASL in the same imaging sessions on the same subjects and the data would share a lot of similarities. Given the time since the above paper was accepted and published, the authors should be able to revise the manuscript accordingly and properly cite that paper. However, there are a significant amount of texts from the above published paper being re-used in this manuscript with minimal changes. For example:

  1. Part 2.1. sample size justification, is almost the same as in Paper 1, except the replacement of ME BOLD with dASL. And there was no justification on halving the effect size of “49%” to “24%”, which is the same number they used in Paper 1.
  2. Part 2.2 study population, is the same as in Paper 1.
  3. Part 2.5 statistical analysis, is almost the same as in Paper 1.
  4. Introduction, paragraph 2, “Structural magnetic resonance imaging (sMRI) has associated meditation …” is the same as in Paper 1, except the last sentence.
  5. Discussion, the last paragraph.

There are more instances and the overall structure of the manuscript (tables/figures) resembles that in Paper 1.

In addition, the authors only mentioned very briefly the findings in Paper 1 regarding the changes of rsFC detected with ME BOLD (different or potential conflicted findings), but the comparison or discussion were minimal (only the paragraph starting line 399 on page 15). A careful comparison and a further interpretation would be helpful to evaluate the validity of the findings for both the ME BOLD and ASL techniques.

There are other issues the authors may consider addressing:

It is known ASL images are subject to noise due to its low SNR, even with background suppression techniques. Could the authors provide sample time series/maps of CBF images, and maybe temporal SNR maps for image quality evaluation?

Dividing the CBF time series by the averaged CBF value removes the quantification aspect of ASL. Does this indicate that flow quantification is not needed? Along the same line, this may indicate that after dividing the averaged CBF value, the authors were observing a “re-distribution” of regional CBF. Would this suggest that larger-scale or global changes of CBF may not be detected? Please discuss.

Line 205. Were there any corrections for multiple comparisons at the cluster-level other than the mentioned p-value of 0.05?

Lines 103-104. Could the authors explain why halving was considered here?

Repeated sentences at lines 233 and 243.

Supp. Figs. 1 and 2. The captions need correction.

Round 2

Reviewer 2 Report

The authors had revised the manuscript following recommendation from reviewers and had successfully addressed most of my concerns. However, there are still some that would need the authors’ further clarification.

Regarding the reused text from previous publication, more effort may be needed. For example,

1) the paragraph starting on line 54: I would recommend the authors to cite the publication on the ME BOLD study and draw connection between the two studies on the same subject.

2) For the study sample and sample size justification. These parts have too much redundant information as the authors had reported this in previous publication. I recommend the authors to significantly shorten these two parts, refer to the recent publication, and only state consideration specific to this study (e.g., the reasoning on halving the effect size analysis regarding ASL vs. BOLD).

Line 263, “SPM8 GM template”. Is the template SPM version dependent? Please correct.

Regarding motion. Line 443: “are more robust to physiological noises and subject motion.” This is inaccurate regarding motion sensitivity. ASL is known to be more sensitive to subject motion due to subtraction, and in this particular study, due to interleaved acquisition. And the motion artefacts may not be readily corrected using conventional MC methods, especially with good background suppression. Please revise or explain. Similarly, the statement in lines 87-89 needs revision.

Line 455-456. Please provide more explanation on the relationship between the polarity difference and the signal source difference.
